# The Use of Audio Signals for Detecting COVID-19: A Systematic Review

**DOI:** 10.3390/s22218114

**Published:** 2022-10-23

**Authors:** José Gómez Aleixandre, Mohamed Elgendi, Carlo Menon

**Affiliations:** 1Biomedical and Mobile Health Technology Lab, ETH Zurich, 8008 Zurich, Switzerland; 2Department of Physics, ETH Zurich, 8093 Zurich, Switzerland

**Keywords:** automatic COVID-19 diagnosis, remote health monitoring, contactless health monitoring, digital health, audio-based health assessment, affordable healthcare

## Abstract

A systematic review on the topic of automatic detection of COVID-19 using audio signals was performed. A total of 48 papers were obtained after screening 659 records identified in the PubMed, IEEE Xplore, Embase, and Google Scholar databases. The reviewed studies employ a mixture of open-access and self-collected datasets. Because COVID-19 has only recently been investigated, there is a limited amount of available data. Most of the data are crowdsourced, which motivated a detailed study of the various pre-processing techniques used by the reviewed studies. Although 13 of the 48 identified papers show promising results, several have been performed with small-scale datasets (<200). Among those papers, convolutional neural networks and support vector machine algorithms were the best-performing methods. The analysis of the extracted features showed that Mel-frequency cepstral coefficients and zero-crossing rate continue to be the most popular choices. Less common alternatives, such as non-linear features, have also been proven to be effective. The reported values for sensitivity range from 65.0% to 99.8% and those for accuracy from 59.0% to 99.8%.

## 1. Introduction

COVID-19 is an infectious respiratory disease caused by the SARS-CoV-2 virus [1]. Soon after its outbreak, near the end of 2019, the disease became a global pandemic. As of 27 February 2022, confirmed cases have surpassed 433 million, with more than 5.9 million reported deaths worldwide [2]. Several variants of the virus have been identified within two years, and some have spread rapidly across the globe. The SARS-CoV-2 virus has strong mutation capabilities, albeit its variants differ in transmissibility and severity. Factors such as advanced age or the presence of comorbidities seem to elevate the risk of developing severe clinical symptoms regardless of the virus variant. Due to the very high transmissibility of SARS-CoV-2, which has already pushed several national healthcare systems to the limit, a continuously increasing amount of effort has been exerted into reducing and controlling the spread of the virus. Individual and collective measures, such as face-mask mandates, the development of vaccines, social distancing, isolation and quarantine periods, and restrictions placed on mobility and public life, have been employed with different degrees of effectiveness [3,4,5].

A prompt diagnosis of newly confirmed cases is critical to the strategy used to fight COVID-19. Since the early days of the pandemic, the gold standard for COVID-19 diagnosis has been the polymerase chain reaction (PCR) test. However, these tests have deficiencies: non-specific amplifications might arise and the sample amount is very limited. Furthermore, they are expensive, slow, and complicated to perform, and require highly qualified professionals and specialized lab equipment [6]. Antigen tests are an alternative based on the detection of viral proteins located on the surface of the virus [6]. These types of tests are more affordable than PCR tests, can be self-performed, and yield results within 15 to 30 min. Nonetheless, they report a poor sensitivity [7,8], which led the WHO to recommend against their use for patient care [9].

Both PCR and antigen tests have the additional burden of being invasive. To overcome these issues, many experts have tried to develop non-invasive, automated, fast testing methods that can be self-performed and offer reliably good results. These works have been based on the analysis of audio signals because COVID-19 generally attacks both the upper and the lower respiratory tracts, with the lungs being particularly affected [10]. Because these are the main body parts involved in the mechanisms of sound production [11], it is safe to assume that COVID-19 might alter the signals of breathing, coughing, and vocalization. Coughing seems to be a common factor in a majority of COVID-19 patients, no matter the variant of the disease [12]. Hence, many studies have used coughing as their data to perform a diagnosis. Such non-invasive, automated methods were also employed before the outbreak of this pandemic. One of the earliest works on automatic cough detection by Matos et al. dates back to 2006 [13], while Shin et al. employed these types of signals for the first time to determine anomalous health conditions [14]. A large number of studies with a similar purpose have been focused on several other illnesses, including pneumonia, pertussis, and asthma. The generally adopted methodology relies on the extraction of spectral and statistical features from the audio signal, which are then fed into a classification algorithm.

Several papers have been published since the pandemic on this topic. Udugama et al. [15] and Thompson et al. [16] focus their reviews on clinical diagnostic methods. Serrurier et al. [17] presented a work that reviews automatic cough acquisition, detection, and classification methods. However, the included studies were not necessarily related to COVID-19. Schuller et al. [18] analyzed the applicability and limitations of computerized audio tools to contain the crisis due to the SARS-CoV-2 virus. A recent publication [19] discussed the challenges and opportunities of deep learning for cough-based COVID-19 diagnosis from an overall perspective. However, this review performs a detailed step-wise examination of the methodology of existing works, either using machine learning- or deep learning-based methods. Specifically, this work focuses on the audio pre-processing and feature extraction processes, which we believe will allow future researchers to improve the performance and generalization of these methods.

## 2. Methods

### 2.1. Study Guidelines

This review was conducted according to the Preferred Reporting Items for Systematic Reviews and Meta-Analyses statement (PRISMA) [20]. A review protocol was drafted using the Preferred Reporting Items for Systematic Reviews and Meta-Analyses Protocols [21].

### 2.2. Search Strategy and Study Eligibility

In this work, the following search terms were employed: (automatic (detection OR diagnosis)) AND (cough OR COVID-19) AND (audio OR sound). To ensure the thoroughness of the review, several databases were searched for papers published between 1 December 2019 and 1 January 2022. Gray literature (e.g., government reports or white papers) was not included in this review in an attempt to only include peer-reviewed studies. The search for this review was completed in April 2022.

### 2.3. Inclusion and Exclusion Criteria

In the first pre-screening phase, duplicate records and papers published in languages other than English, or that were not accessible, were discarded. The screening phase consisted of several steps. First, all the papers that were reviews, case reports, or corrigenda were removed. Then, all the studies that were considered not relevant based on their title were discarded. Finally, the abstracts of the papers were reviewed, the key information was extracted from the main text, and those considered irrelevant were discarded. Reference lists of eligible studies were also hand-searched but no additional studies were included on this basis.

### 2.4. Data Extraction and Risk of Bias

Each potential study for inclusion underwent full-text screening and was assessed to extract study-specific information and data. For each of the included articles, we extracted information from the below perspectives: the year the paper was published, author(s), used dataset(s), devices, number of subjects, subjects’ gender, pre-processing techniques, feature extraction techniques, and evaluation metrics. The authors (JA and ME) relied on QUADAS-2 to assess the applicability of each study, and the authors reached a consensus when needed.

## 3. Results

### 3.1. Study Selection

The paper presents the findings from a systematic review of articles on the topic of *“Automatic COVID-19 detection using audio signals”*. In this section, we present our findings regarding the selection of studies, the employed datasets, the pre-processing and feature extraction stages, and the choice of classifier algorithm (Table 1). The research papers included in this review were found by searching the following databases: PubMed, IEEE Xplore, Embase, and Google Scholar. By applying the aforementioned search strategy, n=659 papers were identified. During the pre-screening phase, 128 papers were discarded, thus leaving n=531 for the screening process. The first stage of this process resulted in the exclusion of 80 studies. As a result of the second screening stage, 233 papers were further removed, thus leaving n=218 papers for eligibility assessment. During the last stage, 170 papers were considered to be unsuitable for the review. Finally, the remaining n=48 papers were selected and included in this review. A flow chart summarizing the selection process is shown in Figure 1.

The majority of the forty-eight studies reviewed in this paper used already existent datasets, while six papers complemented this with their own data, and nine papers exclusively used self-collected audios. The largest available datasets were collected by the Coughvid Project [22] and the University of Cambridge [23], respectively. The former is open-access, whereas the latter can be made available upon request for research purposes. The Coswara [24] and Virufy [25] datasets were widely used among the reviewed papers. The mid-sized Coswara dataset as well as the Coughvid and Cambridge datasets consist of crowdsourced data. The Virufy dataset has both crowdsourced and clinical data. However, only the latter, which amounts to less than 100 files, is accessible. Other small datasets that are not publicly available include the Sarcos [26] and NoCoCoDa [27] datasets. Furthermore, early pandemic studies complemented their data with cough audio files from non-COVID-related datasets. Figure 2 shows the number of papers that employed each of these datasets. The audio files contain breathing, voiced, and principally cough sounds. Although most reviewed studies use cough sounds, other studies combined these with breathing and voiced sounds or chose to use sounds other than coughs. The number of selected samples varies significantly, ranging from 80 [28] to 16,007 [29]. Note that clinically obtained datasets usually require oversight from an institutional review board, which takes time for the studies to run, data to be collected, and then shared. Hence, they exist less often. On the contrary, the applications developed to collect crowdsourced data include disclaimers informing the user beforehand that their audio recordings will be used for research purposes.

### 3.2. Pre-Processing Methods

Pre-processing is generally considered a crucial step in any process involving signal analysis. Employing low-quality data could induce inaccurate results. Hence, to ensure that the analysis and classification results are robust and accurate, pre-processing the raw data is required [30]. While most papers employed several pre-processing techniques, three papers were found [31,32,33], which did not report using any pre-processing steps. Moreover, two other papers [34,35] explicitly specified working with the unprocessed signal. Note that this review does not consider data augmentation techniques to be signal pre-processing, even if some papers include it in their pre-processing sections. Figure 3 presents a summary of the pre-processing techniques used in the reviewed studies.

Cough Automatic Detection (CAD) is an algorithm-based method to detect whether one or more coughs are present in an audio signal. This method was used by 23% of the studies and it is particularly relevant for crowdsourced data. Since audio files recorded in an uncontrolled environment could include unwanted or irrelevant sounds, it could lead to faulty models. The creators of the Coughvid dataset provided each file with a cough probability score based on the result of a self-developed Extreme Gradient Boosting cough classifier [22] and created a cough segmentation algorithm [36]. The combination of these two steps as CAD was used by nine out of the 10 studies that employed the Coughvid dataset. A few studies used other classifiers to detect the presence of cough. Zhang et al. [37] employed a self-designed Convolutional Neural Network (CNN). Tena et al. [38] made use of the TensorFlow-based YAMNet deep neural network. Kamble et al. [39] built a Recurrent Neural Network (RNN) with a long short-term memory architecture. Feng et al. [40] achieved this with a simple k-Nearest Neighbors (k-NN) algorithm. In most cases, these cough detection methods were applied to self-collected data since other datasets do not require this step. In the Coswara and Cambridge datasets, the presence of cough in each audio file has been annotated manually [23,24]; the available recordings from the Virufy dataset were collected in a controlled environment, thus ensuring 100% the presence of cough [25].

Other straightforward pre-processing techniques include re-sampling, amplitude normalization, and manual trimming of the audio signals. These were employed by 35%, 19%, and 39% of the studies, respectively. The trimming procedure was used in two ways: either for silence removal or to segment analyzable sounds into chunks of fixed length. Some papers used a non-manual silence removal process. Feng et al. [40] used a Support Vector Machine (SVM) model to classify the frames of a recording between silence and sound based on the energy of the signal. Pahar et al. [26,41] removed the silences using a simple energy detector, although they did not provide any more details. A similar approach was used by Milani et al. [42], again without specifying the details of the process. Grant et al. [43], Nellore et al. [44], and Sangle and Gaikwad [45] used the signal’s energy as a reference. However, these studies effectively removed the silence parts by hard thresholding. Rao et al. [46] chose a simpler way; they considered any part of the recording below an amplitude threshold to be silent and limited the non-silence parts to a fixed-length duration. Gupta et al. [47] applied a low-pass fourth-order Butterworth filter with a cut-off frequency of 6 kHz (cf. [48]). Kahn et al. [49] removed the DC component of the signal. Unfortunately, they did not further clarify how this was carried out. Noise reduction is another pre-processing technique employed by 12% of the reviewed studies. Tawfik et al. [31] used a wavelet-based de-noising algorithm from the scikit-image library. Zhang et al. [37] chose a spectral gating algorithm based on a hard threshold applied to the Root Mean Square (RMS) of the original signal. Verde et al. [50] used a band-stop filter, as shown in [51]. Three other papers [52,53,54] mentioned noise reduction techniques without specifying how it was performed.

### 3.3. Feature Extraction

A total of 54 extracted feature types were found, which can be classified into four categories: frequency-space (F-type), time-space (T-type), frequency-time-space (FT-type), and statistical (S-type) features. The first three categories refer to features extracted from a frequency, time, and frequency-time representation of the audio signal. The fourth category consists of the features gained from statistical considerations of the corresponding independent variable. The usage of the different features and feature types in the reviewed papers is shown in Figure 4a–d.

The most commonly employed feature was the Mel-Frequency Cepstral Coefficient (MFCC), which is an F-type feature. MFCC features were used in 62% of the studies, although the number of extracted coefficients varied extensively among the papers. Five of the studies [32,37,40,50,54] did not report the number of coefficients that were extracted, and two papers [29,55] mentioned the use of over 60 coefficients. In most cases, 12 or 13 coefficients were used. Other F-type features employed by several papers (10–20%) were the spectral centroid, the spectral bandwidth, the spectral roll-off, the Log-Mel Spectrogram (LMS), and the Chroma Vector (CV). Similar to the MFCC, the CV is a multi-coefficient feature. Two papers [31,52] did not report the number of CV coefficients, while the rest extracted 12 of them. The spectral entropy, the power spectrum density, and the filter bank coefficients were present in 6% of the studies. Other F-type features include the spectral flux, the harmonics-to-noise ratio, the spectral flatness, the spectral contrast, the Teager energy cepstral coefficients, and the linear predictive coding coefficients. These were found in 4% of the papers. Finally, the rest of the F-type features, only observed in 2% of the papers, were spectral spread, relative spectra perceptual linear prediction, Harmonic Ratio (HR), spectral energy, spectral information, noise-to-harmonics ratio, cepstral peak prominence, tonal centroid, non-negative matrix factorization coefficients, eigenvalue ratios, and gamma-tone cepstral coefficients. The formal definitions of these features or their algorithms can easily be found in the signal processing literature. For a formal definition of HR, see [56].

The most frequently evaluated T-type feature was the zero-crossing rate, which was found in 31% of the studies. The fundamental frequency and the RMS were found in 12% and 14% of the papers, respectively. The entropies of the energy, the log-energy, and the formant frequencies were also considered in 6% of the cases. The following T-type features were found in approximately 4% of the studies: the signal’s energy, symbolic recurrence quantification analysis, crest factor, jitter, shimmer, Hjorth descriptors, number and degree of voice breaks, Higuchi and Katz fractal dimension, and maximal phonation time.

Moreover, the S-type features skewness, kurtosis, and autocorrelation were considered by 6%, 12%, and 2% of the papers, respectively. Regarding the FT-type features, the use of non-linear entropies was found in 4% of the studies, while the mean, maximum, and average frequency (cf. [38]), the time-frequency moment, the onset, and some non-linear features (cf. [35]) were found in 2% of the studies.

### 3.4. Classification Algorithms

We identified 13 types of classification algorithms for COVID-19 diagnosis, which were used either independently, i.e., without combining them with other algorithms, or as an ensemble. They belong to two categories: neural network-based and supervised machine learning algorithms. The results are summarized in Figure 5. CNNs were used independently by 29% of the studies, making them the preferred choice amongst the neural network-based algorithms. Other types of networks include RNNs and Feedforward Neural Networks (FNN), which were used independently by 8% and 6% of the reviewed papers, respectively. Some of the studies integrated these three types into ensembles. Alkhodari and Khandoker [57] and Shen et al. [55] combined a CNN with an RNN, while Sanjeev et al. [52] and Fakhry et al. [58] used an FNN together with a CNN. Son and Lee [59] employed a Dense Neural Network (DNN) in combination with a CNN. Finally, Saha et al. [34] utilized a dense convolutional network, as introduced by Huang et al. [60].

Supervised machine learning algorithms were used independently in most cases. SVM, Random Forests (RF), Gradient Boosting Framework (GBF), and k-NN algorithms were observed in 21%, 12%, 10%, and 6% of the studies, respectively. Khan et al. [49] implemented a Decision Tree (DT), while Milani et al. [42] chose a bagged tree, i.e., a DT with a bootstrapping aggregation meta-algorithm (cf. [61]). Chowdhury et al. [62] used an extremely randomized trees classification algorithm. A different approach was taken by Pancaldi et al. [63], who employed a score-based binary classifier. The only instance of a supervised machine learning algorithm ensemble was found in [47], where Gupta et al. implemented a combination of GBF, DT, RF, and k-NN classifiers.

### 3.5. Metrics

For the analysis of the performance of the reviewed papers, the following evaluation metrics were used: specificity, sensitivity, F-score, accuracy, and Area Under the Curve (AUC). As shown in Figure 6a, only 10% of the studies reported all five values, while most of them included three or four metrics (38% and 31% of the papers, respectively). The works by Kamble et al. [64], Grant et al. [43], and Chaudhari et al. [65] only reported the AUC value. The most-reported metric was sensitivity followed by accuracy (Figure 6b); these appeared in 83% and 73% of the papers, respectively. The AUC value (62%) and specificity (60%) were reported similarly often; the F-score was only mentioned in 48% of the studies.

**Table 1 sensors-22-08114-t001:** Summary of the reviewed papers. Non-reported data are given as “nr”. *Datasets*: T = type of collected data (B: breath, C: cough, V: voice), #= number of selected samples. *Pre-processing*: CAD = Cough Activity Detection. *Features*: Extracted features (number of features, only reported if the given feature is not a one-element category); numbers of features given as (x×3) indicate the use of the fundamental coefficients together with the corresponding Δ and $\Delta^{2}$ values. *Classifier*: employed classification algorithm. *Metrics*: Specificity (SP), Sensitivity (SE), F-score (F1), Accuracy (ACC), and Area Under the Curve (AUC), given in percentages.

Date	Ref.	Datasets [T] (#)	Device (SR)	Pre-Processing	Features (#)	Classifier	SP (%)	SE (%)	F1 (%)	ACC (%)	AUC (%)
2022	Son and Lee [59]	Coughvid [C] (6092)	nr (48 kHz)	CAD, Re-sampling	MFCC (13), SP, LMS, SCn, SB, SC, SR, CV (12)	CNN + DNN	94.0	93.0	n r	94.0	n r
2022	Tawfik et al. [31]	Coswara, Virufy [C] (1171, 121)	nr (nr)	Noise reduction, Trimming, CAD	MFCC (20), O, CV (nr), SC, SR, ZCR, SB, CQT	CNN	99.7	99.6	98.4	98.5	n r
2022	Alkhodari and Khandoker [57]	Coswara [B] (480)	Smartphone (48 kHz)	Re-sampling	MFCC (13), K, SE, ZCR, EE, Sk, HFD, KFD	CNN + RNN	93.0	93.7	93.3	93.3	88.0
2022	Kamble et al. [64]	Coswara [C] (1436)	nr (44.1 kHz)	n r	TECC (63×3)	GBF	n r	n r	n r	n r	86.6
2022	Gupta et al. [47]	Coughvid [C] (2500)	nr (nr)	CAD, Normalization, BF, Re-sampling	MFCC (13), RMS, SR, SC, ZCR, CF	DT + RF + k-NN + GBF	n r	n r	79.9	79.9	79.7
2022	Haritaoglu et al. [29]	Coswara, Virufy, Coughvid, Own data [C] (Total: 16007)	nr (nr)	CAD, Re-sampling	MFCC (64), LMS	SVM, CNN	SVM: 51.0, CNN: 74.0	SVM: 89.0, CNN: 77.0	n r	SVM: 59.0, CNN: 75.0	SVM: 80.7, CNN: 80.2
2022	Islam et al. [56]	Virufy [C] (121)	nr (nr)	Trimming	SC, SE, SF, SR, MFCC (13), CV (12), HR	CNN	100.0	95.0	97.4	97.5	n r
2022	Pancaldi et al. [63]	Own data [B] (224)	Stethoscope (4 kHz)	Trimming	LPCC (8)	BC	81.8	70.6	n r	75.0	n r
2021	Milani et al. [42]	Coswara [V] (nr)	nr (nr)	Silence removal	GTCC (nr)	k-NN, BT	n r	n r	90.9	90.0	n r
2021	Nellore et al. [44]	Coswara [C] (1273)	nr (nr)	Silence removal	FBC (128×3), F ${}_{0}$, F ${}_{n}$ (4), RMS	FNN	34.3	85.6	n r	n r	65.7
2021	Pahar et al. [41]	Coswara, Cambridge, Sarcos [C] (1171, 517, 44)	nr (Cos.: 44.1 kHz, Cam.: 16 kHz, Sar.: 44.1 kHz)	Re-sampling, Silence removal	ZCR, FBC, K, MFCC (13×3)	CNN	92.7	95.7	n r	94.0	95.9
2021	Rahman and Lestari [66]	Coswara, Coughvid [C] (5120, 2601)	nr (nr)	Cou.: CAD, Cos.: None	NMFC	SVM	55.6	90.0	n r	n r	73.3
2021	Saha et al. [34]	Coswara [C] (426)	nr (44.1 kHz)	None	LMS	DCN	98.8	99.5	n r	99.5	98.9
2021	Sangle and Gaikwad [45]	Coswara [C] (825)	nr (44.1 kHz)	Normalization, Silence removal	MFCC (13×3), ZCR, K, Sk	CNN	99.2	92.3	96.0	96.0	96.0
2021	Sanjeev et al. [52]	Own data [C] (1350)	nr (nr)	Noise reduction, Silence removal	MFCC (20), SC, CV (nr), RMS, ZCR, SB, SR	FNN + CNN	n r	82.0	83.0	85.0	94.0
2021	Shkanov et al. [67]	Own data [C] (1876)	nr (nr)	Trimming	EVR (2)	SVM, RF	n r	n r	SVM: 93.0, RF: 94.0	SVM: 93.0, RF: 94.0	n r
2021	Wang et al. [68]	Own data [B] (104)	Stethoscope (nr)	Trimming	LMS	CNN	93.1	90.0	n r	91.3	n r
2021	Solak [35]	Virufy [C] (194)	nr (48 kHz)	None	NLF (9)	SVM	96.8	93.1	n r	95.8	93.2
2021	Zhang et al. [37]	Own data, Coswara, Virufy [C] (321, nr, nr)	nr (nr)	Noise reduction, CAD, Trimming	MFCC (nr)	CNN	95.8	95.3	96.1	95.8	98.1
2021	Chowdhury et al. [62]	Cambridge, Coswara, NoCoCoDa, Virufy [C] (525, 1319, 73, 121)	nr (nr)	Re-sampling	MFCC (40), TC (6), LMS, CV (12), SCn (7)	ET, GBF	n r	ET: 74.0, GBF: 80.0	n r	n r	ET: 83.0, GBF: 82.0
2021	Rao et al. [46]	Coughvid, Coswara [C] (nr)	nr (nr)	Silence removal, Re-sampling	LMS	CNN	77.9	80.0	n r	n r	82.3
2021	Shen et al. [55]	Coswara, Own data [C] (610, 707)	nr (nr)	Re-sampling, Trimming	bw-MFCC (128)	CNN + RNN	93.3	81.8	n r	n r	96.1
2021	Tena et al. [38]	Own data, Coswara, Cambridge, Virufy [C] (813 total)	nr (nr)	CAD, Re-sampling, Normalization	SEn, TFM, MxF, MF, AF, NLE (3), SI, K	RF	85.1	86.0	85.6	85.5	89.6
2021	Vahedian-Azimi et al. [32]	Own data [V] (748)	Recorder (44.1 kHz)	n r	F ${}_{0}$, J, Sh, HNR, NHR, AC, CPP, MFCC (nr), MPT, NVB, DVB	FNN	n r	91.6	90.6	89.7	n r
2021	Erdoğan and Narin [53]	Virufy [C] (1187)	nr (nr)	Normalization, Noise reduction	NLE (6)	SVM	97.3	99.5	98.6	98.4	n r
2021	Grant et al. [43]	Coswara [V] (1199)	nr (44.1 kHz)	Silence removal, Trimming, Normalization	MFCC (20×3), RASTA-PLP (20)	RF	n r	n r	n r	n r	79.4
2021	Irawati and Zakaria [69]	Coswara, Virufy [C] (150, 121)	nr (Cos.: 44.1 kHz, Vir.: 48 kHz)	Trimming	MFCC (20), SB, RMS, ZCR	GBF	n r	n r	Cos.: 87.0, Vir.: 82.0	Cos.: 86.0, Vir.: 86.2	n r
2021	Khan et al. [49]	Own data [C] (1579)	Stethoscope (8 kHz)	DC removal, Normalization, Trimming	HD (3)	k-NN, DT	k-NN: 97.8, DT: 92.0	k-NN: 99.8, DT: 93.9	n r	k-NN: 99.8, DT: 94.4	n r
2021	Melek-Manshouri [70]	Virufy [C] (121)	nr (nr)	Trimming	MFCC (13)	SVM	91.7	98.6	n r	95.9	n r
2021	Melek [71]	NoCoCoDa, Virufy [C] (59, 121)	nr (NoC.: 44.1 kHz, Vir.: 48 kHz)	Trimming	MFCC (19)	k-NN	n r	97.2	98.0	98.3	98.6
2021	Nessiem et al. [72]	Cambridge [C, B], (1035)	nr (16 kHz)	Re-sampling	LMS	CNN	62.8	77.6	n r	67.7	77.6
2021	Verde et al. [50]	Coswara [V] (166)	nr (nr)	Noise reduction	MFCC (nr), F ${}_{0}$, J, Sh, SC, SR	RF	70.6	94.1	84.2	82.4	90.1
2021	Banerjee and Nilhani [73]	Coswara, Coughvid [C] (233, 640)	nr (nr)	CAD, Trimming	LMS	CNN	99.7	67.2	79.3	96.0	n r
2021	Kamble et al. [39]	Coswara [C] (1253)	nr (44.1 kHz)	CAD	TECC (40×3)	GBF	53.6	80.5	n r	n r	76.3
2021	Maor et al. [74]	Own data [V] (434)	nr (nr)	Re-sampling	LMS	RF, SVM	53.0	85.0	n r	n r	72.0
2021	Pahar et al. [26]	Coswara, Sarcos [C] (1171, 44)	nr (44.1 kHz)	Silence removal, Normalization	MFCC (13), ZRC, K, LE	CNN, RNN	CNN: 98.0, RNN: 96.0	CNN: 93.0, RNN: 91.0	n r	CNN: 95.3, RNN: 92.9	CNN: 97.6, RNN: 93.8
2021	Vrindavanam et al. [75]	Freesound, Coswara, Cambridge [C] (150 total)	nr (44.1 kHz)	Re-sampling	MFCC (12), ZCR, F ${}_{0}$, RMS, CF, SB, PSD, SC, SR, LE, SP, LPCC, F ${}_{n}$ (4)	SVM, RF	n r	SVM: 81.2, RF: 73.8	SVM: 78.4, RF: 78.0	SVM: 83.9, RF: 85.2	n r
2021	Fakhry et al. [58]	Coughvid [C] (1880)	nr (44.1 kHz)	CAD, Re-sampling	MFCC (13), LMS	CNN + FNN	99.2	85.0	n r	n r	91.0
2021	Han et al. [76]	Own data [V] (828)	Smartphone, Computer (nr)	Trimming, Re-sampling, Normalization	F ${}_{0}$, MFCC (12), RMS, HNR, ZCR	SVM	82.0	68.0	n r	n r	79.0
2021	Khriji et al. [54]	Audioset, ESC-50 [C, B] (nr)	nr (nr)	Noise reduction	PSD, FBC, MFCC (nr)	RNN	n r	78.8	79.0	80.3	n r
2021	Pal and Sankarasubbu [77]	Own data [C], (150)	nr (nr)	Re-sampling, Normalization, Trimming	MFCC (12), LE, EE, ZCR, Sk, F ${}_{n}$ (4), F ${}_{0}$, K	FNN	90.3	90.1	90.6	90.8	n r
2021	Feng et al. [40]	Coswara, Virufy [C] (633, 25)	nr (nr)	Silence removal, CAD	MFCC (nr), E, EE, ZCR, SE, SC, SS, SF	RNN	n r	n r	n r	Cos.: 90.0, Vir.: 81.2	Cos.: 92.8, Vir.: 79.0
2021	Lella and Pja [78]	Cambridge [C, B, V] (256)	Smartphone, Computer (16 kHz)	Trimming	MFCC (159×3)	CNN	n r	n r	93.5	92.3	n r
2021	Brown et al. [23]	Cambridge [C, B] (491)	Smartphone, Computer (nr)	Trimming, Re-sampling	MFCC (13×3), SC, RMS, SR, ZCR	SVM	n r	72.0	n r	n r	82.0
2021	Mouawad et al. [33]	CVD [C] (1927)	nr (22.05 kHz)	n r	s-RQA (nr)	GBF	n r	65.0	62.0	97.0	84.0
2020	Hassan et al. [28]	Own Data [C] (80)	Microphone (nr)	Trimming	SC, SR, ZCR, MFCC (nr×3)	RNN	n r	96.4	97.9	97.0	97.4
2020	Chaudhari et al. [65]	Virufy, Coughvid, Coswara [C] (1442, 941, 105)	Smartphone (nr)	Re-sampling	MFCC (39), LMS	CNN	n r	n r	n r	n r	77.1
2020	Bansal et al. [79]	Audioset, ESC-50 [C] (501 total)	nr (44.1 kHz)	Trimming	MFCC (40), SC, SR, ZCR	CNN	n r	81.0	69.6	70.6	n r

## 4. Discussion

The development of a rapid, self-performable, automatized detection method can be a helpful tool for diagnosing and controlling the spread of diseases. In the case of highly contagious pandemic diseases, this is particularly important. In this systematic review, we selected and analyzed studies that addressed this issue analogously: first, the data from either crowdsourced or clinically obtained datasets underwent different signal processing techniques; then, a set of features was extracted and fed into a classification algorithm (see Figure 7), which characterized the data as positive or negative for COVID-19. In some instances, demographic information or clinical data are also used as input for the classifier.

Most of the employed datasets primarily consist of cough sounds. This type of sound is more suitable for developing a generalizable method. Breathing sounds often need to be recorded with a stethoscope to ensure good quality, which requires a clinical recording environment. Vocalized sounds are less general; some instructions or a detailed description of the kind of sounds that must be generated are required. Additionally, cough sounds have the advantage of being anharmonic. The features of two arbitrarily produced coughs can be easily compared [80], whereas a vocalized sound (e.g., a sustained vowel) is affected by the anatomy, thus making it necessary to differentiate between ages or genders. Furthermore, the features obtained from a recorded speech or phrase could be language-biased due to the associated defining phonetics. Nonetheless, some of the reviewed studies that do not employ cough sounds have reported promising results.

The amount of original data impacts the statistical significance of a study. In the past, data augmentation techniques have been used successfully for developing cough detection algorithms [81]. They have been most often applied to enlarge the available datasets or to balance the majority and minority classes. It has been shown that augmenting a balanced dataset to 4–5 times its original size can improve classification accuracy by more than 10% [82]. We consider that this technique needs to be approached cautiously. While a classifier should perform better when trained with more data, typical data augmentation algorithms create new data points by slightly variating the original data. Such a technique could result in the introduction of some bias. Some studies have investigated the performance of data augmentation in cough detection or image recognition, i.e., when the original pre-processed file is given as input to the classifier. To the best of our knowledge, there is a lack of detailed studies on whether data augmentation results in a positive impact or a negative bias in classification based on extracted features.

The choice of extracted features is essential to ensure effective classifier performance. Both the shortage of meaningful features and the abundance of irrelevant ones can contribute to the deterioration of classification efficiency. For example, in the case of MFCC, the most common choice was between 12 and 13 coefficients. The reasons for this choice might be twofold. While it has been shown that using eight to 14 coefficients can suffice for speech analysis [83], reducing the large number of initially extracted MFCC is often desired. For this purpose, the Karhunen–Loeve transform, which in speech analysis can be approximated by a direct cosine transform [84], is a frequent choice [85]. Several of the proposed methods only employ a subset of the extracted features. Some use a dimensionality reduction algorithm to select an optimal feature subset. We could not find two papers with the same choices regarding feature extraction and selection as well as classification algorithms. Moreover, these choices are not thoroughly justified, and the validation methods employed also varied. Therefore, and based on the results of this review, it is difficult to conduct a quantitative analysis of which features should be extracted, which feature selection algorithm works best, or which classifier is more effective.

Equally important is the quality of the extracted features, which is closely related to the quality of the audio data. Unless data are collected very carefully in a controlled environment, some pre-processing is needed to ensure good quality. Thus, it is safe to assume that self-performed recordings will be noisy and unclean. Noise reduction techniques were infrequently considered in the reviewed studies. This is a surprising fact, given that the majority of the used audio files came from crowdsourced datasets, which had not been curated in regard to noise. Noisy signals could bias the extracted features and obstruct the silence removal and CAD processes.

A coughing sound typically consists of an explosive, an intermediate, and a voiced phase [86]. While some coughs may be multiple (i.e., the next explosive phase starts before the last voiced phase ends), in most cases, the separation between individual coughs is clear. To analyze the sound signal, we also consider full cycles (i.e., all three phases) to be more appropriate than segments of fixed length. Extracting the individual coughs via manual trimming or annotation can be extremely time-consuming when working with large datasets. Additionally, it is not suitable for an automatized, self-performable detection method. The usage of an algorithm-based technique solves this issue, but it can be problematic for crowdsourced data, which do not undergo any screening of the sound types in the recording. By embedding cough segmentation into a CAD algorithm, both problems can be tackled at once.

Re-sampling is a technique that aims to improve computational speed by reducing the number of frames and thus the number of data points of a given audio file. A detailed comparison of the sampling rates and metric values in the reviewed studies did not show any correlation between these factors. In general, the choice of sampling rate should provide a good enough improvement in computational speed without a sensible loss of information. Although most of the studies that considered re-sampling agree with this choice, Alkhodari and Khandoker [57] achieved good results despite reducing the sampling rate from 48 kHz to 4 kHz.

The reported metrics are the most direct way of evaluating the performance of a proposed method. These values are greatly affected by the type and amount of available data and by the pre-processing steps, feature extraction, and classification algorithm that are used. As a consequence of such diversity, comparing the metric results between papers is rather ineffective. Nevertheless, to offer a qualitative evaluation of the performance of the reviewed studies, here, we will discuss the best-performing papers. The entire diagnosis process includes several steps. Thus, there is a large number of variables involved. Each of the metrics describes a different aspect of the performance of a method. Therefore, the quality of a study improves when several metrics are reported since a high value on one metric does not directly imply high values in other ones. In this way, we considered those papers with three or more reported evaluation metrics of 90% or higher each to be amongst the best performing. Note that other criteria could be used to determine the best-performing models. However, evaluating the study’s performance based on these metrics has to be consistently used in all papers.

Son and Lee [59] proposed concatenating the outputs of a CNN, fed with a handcrafted feature set, and a DNN that receives the LMS as input. While they did not use any cross-validation method, they used 70% of the data for training, 15% for validation, and 15% for testing. Although they accessed several datasets, they only reported the results of their proposed method with the Coughvid dataset. Furthermore, they worked with a heavily unbalanced dataset containing 10 healthy subjects for each COVID-19-positive subject (10N to 1P). A similar approach was taken by Tawfik et al. [31]. They mixed two different datasets but only employed a slightly unbalanced subset (2N to 1P). The evaluation was done using a 10-fold cross-validation technique. These two differences might explain why they obtained better results than Son and Lee [59]. Cross-validation improves the robustness of the method, while the use of balanced datasets for the training phase is particularly important, even if it does not play a crucial role in the testing phase [87].

Islam et al. [56] reported excellent results. However, that study used a balanced but very small dataset, which was clinically collected, i.e., noise-free and annotated. They used a 70/20/10 split of their data and used five-fold cross-validation. Their pre-processing techniques were tailored for this type of data, thus making them less generalizable. A more detailed explanation of their feature extraction process would be desired. Islam et al. [56]) compared features extracted from COVID-19 positive subjects and healthy subjects. However, silent frames, which were not removed before the feature extraction, can be easily identified. That might have biased the results of their work. Two other studies employed the same dataset, and while they took different approaches, they achieved similar results. Melek-Manshouri [70] extracted MFCC from the pre-processed signal and selected the best ones using a sequential forward search method, whose resulting output was fed into an SVM with an RBF kernel. The evaluation was performed with a leave-one-out cross-validation technique. Melek [71] employed a slightly larger but still small dataset and used a Euclidean k-NN classifier with leave-one-out cross-validation to determine the hyperparameters for feature extraction and evaluate their proposed method. In the case of the k-NN classifier, feature selection using a sequential forward search technique did not improve the results. However, the study disregarded noisy recordings. Such a treatment is not generalizable since only recordings collected in a controlled environment, such as a clinic or hospital, can be assuredly noise-free.

Saha et al. [34] performed data augmentation by pairing the original signals with random white noise and time-stretching the resulting audio. Consequently, the amount of initial data was enlarged 40-fold. Although data augmentation can help improve the classifier performance, as discussed above, Saha et al. [34] failed to address the main issue: their dataset was deeply unbalanced (10P to 1N). The evaluation was done with a 70/10/20 data split but without cross-validation. Other studies using data augmentation techniques are those by Sangle and Gaikwad [45] and Solak [35]. In both cases, oversampling was employed to balance the datasets. The former did an 80/20 split for evaluation without cross-validation. They also tried different sets of handcrafted features. However, their results seemed inconclusive since the accuracy and F-score values for the various combinations suggested random behavior. The dataset employed by the latter was still very limited and the evaluation techniques were not reported. Interestingly, Solak [35] worked with a set of handcrafted, nonlinear features motivated by the nonlinear nature of cough signals.

Erdoğan and Narin [53] also employed non-linear features. They had access to the complete Virufy dataset, including a large corpus of crowdsourced data. Unfortunately, since these data are not open-access, the replicability of this study is strongly hindered. On a positive note, the dataset was highly balanced, with almost the same number of annotated recordings from COVID-19 subjects and healthy subjects. They used the ReliefF algorithm to select the best among the extracted features. The evaluation was performed using an SVM classifier with a linear kernel and five-fold cross-validation.

Pahar et al. [26] explored two crowdsourced datasets: the large Coswara dataset and the small Sarcos dataset. After balancing via the synthetic minority over-sampling technique [88], the former was used to train both a CNN and an RNN model. The RNN model, in combination with leave-p-out cross-validation and sequential forward search techniques, was first applied to select the best subset of extracted handcrafted features. Both classifiers were subsequently tested while employing the same cross-validation technique. The CNN model had the best results when using the Coswara dataset for evaluation, whereas the RNN performed best with the Sarcos dataset. Moreover, the selected hyperparameters were different in both cases. Some of the results suggest that their proposed methods might not be generalizable. For example, when they applied the CNN classifier with its corresponding hyperparameters to the Sarcos dataset, the accuracy and AUC values decreased by 20%, and the specificity was 40% lower. The same authors took a different approach in another study [41]. First, they pre-trained a CNN model with several not-labeled datasets, which were created to develop cough detection or tuberculosis diagnosis algorithms. This technique is called transfer learning and it serves the same purpose as data augmentation. Although it allows the neural network to be trained with more original data, it might introduce some bias to the extracted features. For evaluation, the data was split (80% for training and 20% for testing), and nested cross-validation was applied.

Zhang et al. [37] worked with a small, self-collected dataset complemented with crowdsourced data. Unfortunately, they failed to report the amount of data used from each crowdsourced corpus. For their self-collected data, they applied a CAD pre-processing method. That method needs manual annotation, thus hindering the generalizability of their model. Furthermore, they used MFCCs as features but they did not mention the number of extracted coefficients. The entire dataset with an 80/10/10 split was used for training and testing their CNN-based CAD algorithm. The evaluation was performed with a five-fold cross-validation technique, although they only considered a small data subset.

The studies by Wang et al. [68], Pal and Sankarasubbu [77], and Hassan et al. [28] used very small datasets and failed to report whether a cross-validation technique was used. In Wang et al. [68], the data consisted of breath sounds collected with a stethoscope. That study applied data augmentation in the same way as Saha et al. [34], but did not mention how many new data points were created. Hassan et al. [28] used the smallest dataset of all the reviewed papers. Despite the imbalance (3N to 1P), neither data augmentation nor balancing techniques were applied.

The work by Khan et al. [49] differs from others in that they employed self-collected cough sounds from lung auscultations. They also trimmed the signal in frequency space, only considering the parts with frequencies up to 2 kHz. While this type of data can be high quality, it fails to address the desired self-performability criteria. Additionally, it leaves unanswered the question of whether the same method would offer as good results using other data types. They used Hjorth descriptors as features, which was not observed in the other reviewed papers. They evaluated their method with both a k-NN (k = 10) and a DT classifier, but they failed to report the validation technique they employed. In this case, the classification was three-fold and included COVID-19 positive, dry cough, and wet cough.

The reviewed studies work towards the development of an automatized, self- performable COVID-19 detection method using audio signals. We consider that cough signals are optimal for this purpose. They can be easily produced and recorded and are less biased by demographic factors. To improve the generalizability of the results, it is essential to have large amounts of data. Despite showing good performance metrics, several of the reviewed studies were performed with small (<200) datasets. Techniques such as data augmentation and transfer learning can help increase the amount of available data, but we strongly encourage merging all the available open-access datasets. Therefore, the development of a comprehensive pre-processing algorithm that allows combining data collected in different environments is considered to be decisive. In particular, noise reduction and segmentation techniques can improve the quality of the raw data. For non-annotated datasets, we recommend the use of the CAD algorithm developed by the authors of the Coughvid dataset. When applied together with a cough segmentation algorithm, it yields good results, as shown by Son and Lee [59] and Fakhry et al. [58]. Re-sampling could also be beneficial, especially when employing large datasets. A sampling rate between 16 and 22 kHz is regarded as an optimal choice that enhances computational speed without an extensive loss of information. Particular emphasis should be put into the feature extraction process. We believe that a detailed study of each individual feature and its potential to characterize an audio signal as COVID-19 positive or negative would lead to more informed feature choices. In addition to the common feature types used in audio signal analysis, alternative options, as shown by several in several of the reviewed papers, can offer promising results as well. A better understanding of an optimal set of features for the particular case of COVID-19 characterization could also create the opportunity to conduct a more detailed study of different classification algorithms.

Based on the results of this review, we believe that the following recommendations could lead to better and, in particular, more generalizable results:Database creators should aim to collect diverse data, including a wide range of ages, genders, ethnicities, etc. These datasets need also be publicly available since combining several databases would allow researchers to have a larger data corpus and reduce the possible biases introduced by demographic factors.The pre-processing methods need to be automated and be as non-data-specific as possible. It is essential that researchers consider carefully whether re-sampling techniques improve computational speed enough to compensate for the information loss. An efficient CAD algorithm is critical since manual trimming and labeling become highly impractical when working with large datasets. Data augmentation should be avoided in favor of precise cough segmentation techniques and the creation of large datasets.The impact of each feature needs to be investigated individually, thus increasing the efficiency of manual feature extraction processes, allowing the adaptation of the neural network architectures, and improving the generalizability of the methods. Moreover, by fixing a subset of features that does not depend on the available dataset, researchers could design studies to test the performance of several machine learning and deep learning models as the only dependent variable.

## 5. Conclusions

Over the last two decades, the application of neural networks and machine learning algorithms for cough detection, cough discrimination, and the diagnosis of respiratory diseases, such as pneumonia or asthma, via the analysis of audio signals has notably progressed. Since the outbreak of the COVID-19 pandemic, all efforts have been focused on implementing and improving these methods to diagnose this disease. This systematic review has shown that state-of-the-art studies offer promising results. However, an important issue is the generalizability of the proposed methods. The scientific community should aim to develop a pre-screening tool that could be used instead of the usual test methods (PCR and antigen tests). Such a solution would be fast, accurate, and self-performable (without the necessity of specialized equipment or knowledge) and would significantly reduce the environmental impact associated with mass antigen and PCR tests. The first step is the creation of large open-access datasets of audio recordings that would allow many scientists to contribute to that objective. In combination with this, the development of general, non-dataset-specific pre-processing algorithms and a more detailed study of how both conventional and uncommon acoustical and statistical features can differentiate between healthy and infected subjects should be targeted. In particular, we suggest the individual study of every feature and the development of a quantifying method to establish the adequacy of each one for distinguishing between healthy and COVID-19-positive subjects. Although this might seem similar to dimensionality reduction methods, such as principal component analysis, an individual study of each feature would minimize the correlation in the feature selection step, thus strongly contributing to the reduction of error sources as well as to the generalization of the method.

## Figures and Tables

**Figure 1 sensors-22-08114-f001:**
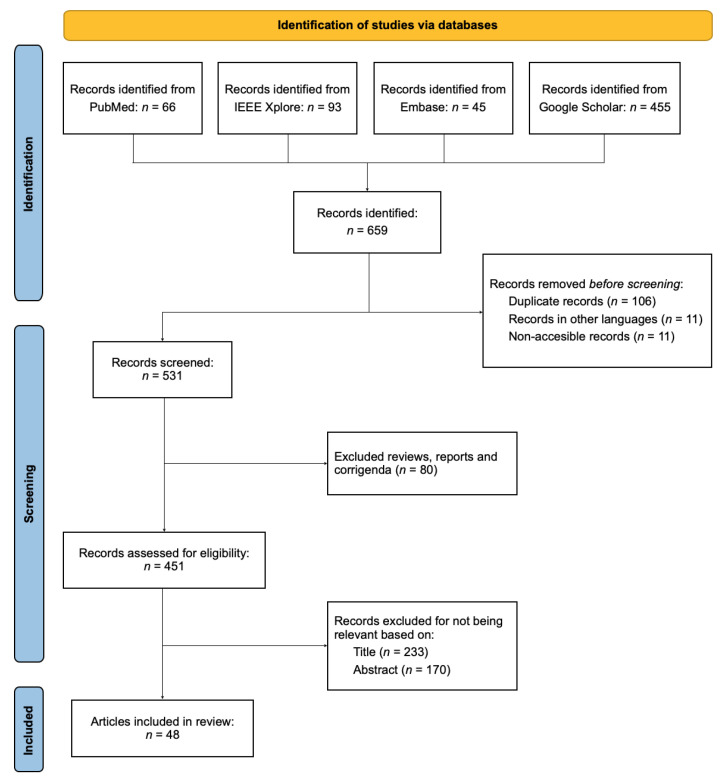
Flowchart summarizing the search and selection process of the reviewed papers.

**Figure 2 sensors-22-08114-f002:**
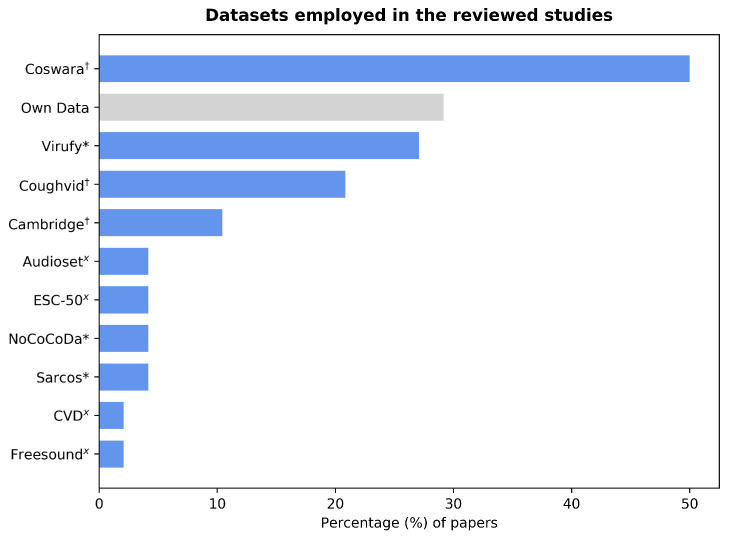
Bar graph showing the percentage of papers that used each of the identified datasets. Of them, 29% employed their own self-collected data, which have not been formally turned into a dataset. Those marked with “x” are non-COVID datasets that originated before the start of the pandemic; those marked with “*” include clinically collected data and those marked with “†” consist of crowdsourced data.

**Figure 3 sensors-22-08114-f003:**
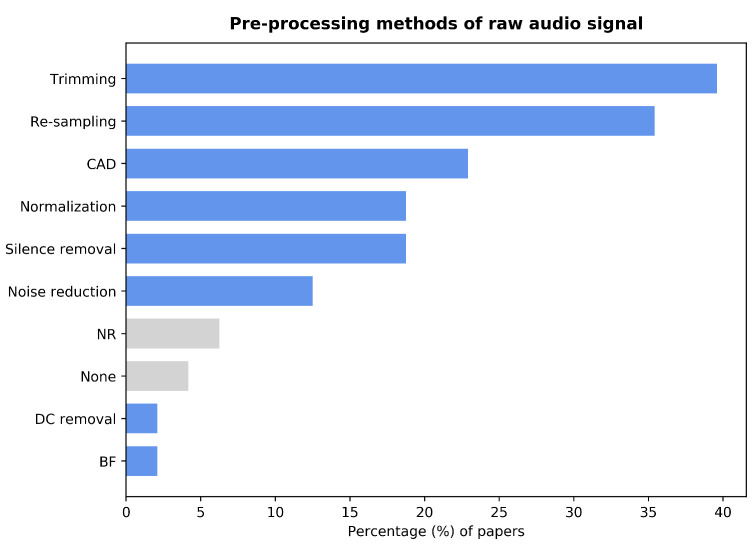
Bar graph showing the percentage of papers that employ each of the pre-processing techniques. *Trimming*: manual segmentation of the raw audio signal. *Re-sampling*: reduction of the sampling rate of the raw audio signal. *CAD*: algorithm-based Cough Automatic Detection method. *Normalization*: amplitude normalization of the raw audio signal. *BF*: application of a low-pass Butterworth filter. *Silence removal*: algorithm-based method to remove silence parts of the raw audio signal. *DC removal*: subtraction of the DC part of the raw audio signal. *None*: papers that report using the raw audio signal for feature extraction. *NR*: papers that did not report any pre-processing.

**Figure 4 sensors-22-08114-f004:**
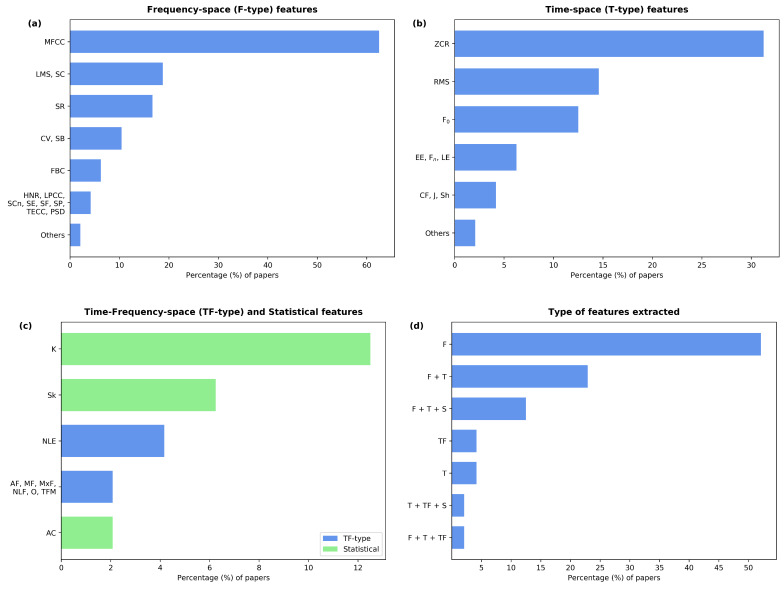
Percentage of papers that extracted each listed feature grouped according to feature types: F-type (**a**), T-type (**b**), and TF-type and S-type (**c**). Additionally, the percentage of papers using each of the given combinations of feature types is shown (**d**). **Legend** ⇒ Mel-Frequency Cepstral Coefficients (MFCC), Log-Mel Spectrogram (LMS), Chroma Vector (CV), Spectral Centroid (SC), Spectral Roll-off (SR), Spectral Bandwidth (SB), Filter Bank Coefficients (FBC), Harmonics-to-Noise Ratio (HNR), Linear-Predictive Coding Coefficients (LPCC), Spectral Contrast (SCn), Spectral Entropy (SE), Spectral Flux (SF), Spectral flatness (SP), Teager Energy Cepstral Coefficients (TECC), Power Spectrum Density (PSD), Zero Crossing Rate (ZCR), Root-Mean-Square (RMS), fundamental Frequency (F${}_{0}$), Entropy of the Energy (EE), formant Frequencies (F${}_{n}$), Log Energy (LE), Crest Factor (CF), Jitter (J), Shimmer (Sh), Kurtosis (K), Skewness (Sk), Non-Linear Entropies (NLE), Average, Mean and Maximum Frequency (AF, MF, and MxF), Non-Linear Features (NLF), Onset (O), Time-Frequency Moment (TFM), and AutoCorrelation (AC).

**Figure 5 sensors-22-08114-f005:**
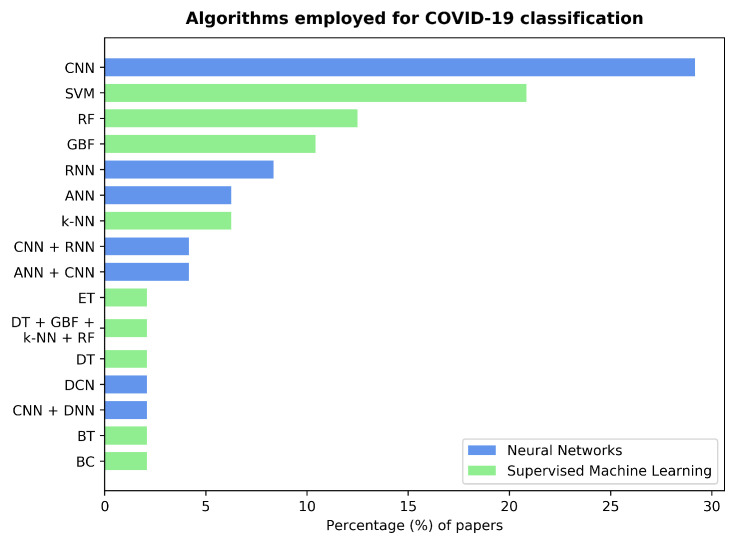
Bar graph showing the percentage of papers employing different classifier algorithms or ensembles. The colors of the bars highlight the type of algorithm, with 54% of the papers using neural network-based algorithms and 48% of them using supervised machine learning. Note that some studies report results with several algorithms (see Table 1); thus, the sum of these percentages exceeds 100%. **Legend**⇒ Convolutional Neural Network (CNN), Support Vector Machine (SVM), Random Forests (RF), Gradient Boosting Framework (GBF), Recurrent Neural Network (RNN), Feedforward Neural Network (FNN), k-Nearest Neighbors (k-NN), Extremely randomized Trees (ET), Decision Tree (DT), Dense Convolutional Network (DCN), Dense Neural Network (DNN), Bagged Tree (BT), and Binary Classifier (BC).

**Figure 6 sensors-22-08114-f006:**
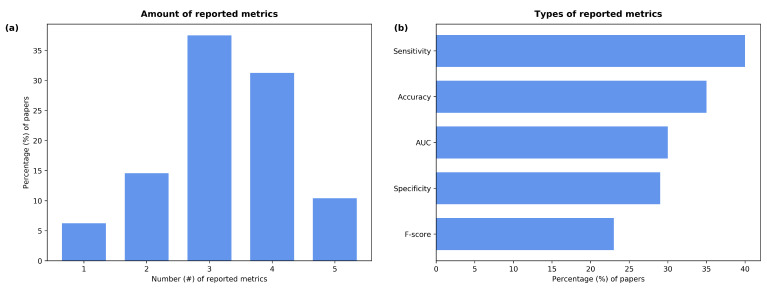
Percentage of papers that (**a**) reported the given number of metrics and (**b**) used each one of them.

**Figure 7 sensors-22-08114-f007:**
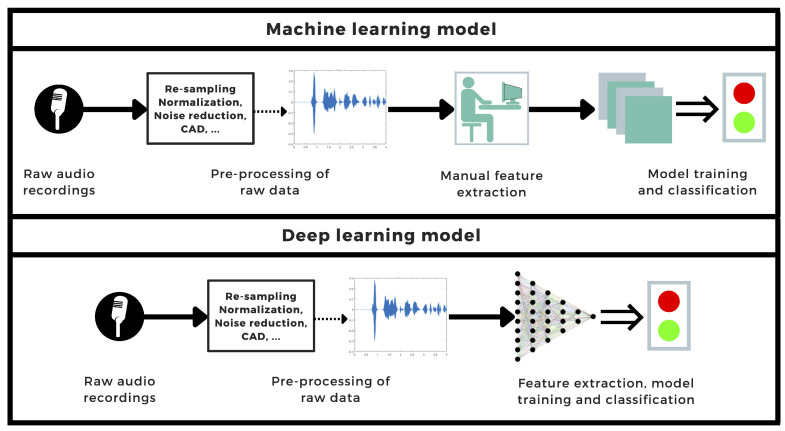
General workflow diagram of common machine learning- and deep learning-based COVID-19 detection processes using audio signals.

## Data Availability

Not applicable.

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
