# Peer review of "The Use of Audio Signals for Detecting COVID-19: A Systematic Review"

_sensors, 2022, doi:10.3390/s22218114_

Round 1
Reviewer 1 Report
This well-done literature review provided a sound strategy for identifying datasets available. Overall clearly written work that is scientifically rigorous.
A few minor points for the authors to consider.
1. The search term "(automatic [detection OR diagnosis]) AND (cough OR COVID-19) AND (audio OR sound)" produces an error in PubMed with the bracketed term [...]. It is recommended the authors advise on this issue.
2. It will help some readers to clarify examples fo the term "gray literature" that are working papers, government documents, white papers and evaluations. Were legitimate government reports eliminated?
3. How was low-quality established? Publication impact factor? Other?
4. It may be obvious to some readers that a crowdsourced data set is more of an open-access strategy relative to formalized clinical trial work, but the authors may wish to explain the data set as IRB-exempt participatory method of building a dataset.
5.Figure 3 is unique and an excellent addition to the manuscript. Same comment for Figure 5, but Figure 4 was not clear; not sure if that is a font or page-size limitation.
6. Table 1 reflects a significant amount of work, but could this be simply a data appendix available on-line? There were formatting issue on the PDF reviewed and the amount of data is overwhelming to include in the paper itself.
7. Lines 249-259 are informative and critical. Were databases segregated on method of data collection?
9. Lines 315-320 are important as they suggest a way to identify best-performing paper. Please consider expanding the narrative for clarity.
10. The conclusion is somewhat anti-climatic. Possibly stronger recommendations based on the very comprehensive study? The reader si left without a clear sense of the authors' recommendation on what others might do (specifically) to improve acoustic detection of COVID-19.
Author Response
This well-done literature review provided a sound strategy for identifying datasets available. Overall clearly written work that is scientifically rigorous.
Author response: We thank the reviewer for the overall positive feedback for recognizing the merit of our work.
- The search term "(automatic [detection OR diagnosis]) AND (cough OR COVID-19) AND (audio OR sound)" produces an error in PubMed with the bracketed term [...]. It is recommended the authors advise on this issue.
Author response: We thank the reviewer for pointing out this formatting detail.
Author action: The square brackets have been changed to round brackets.
- It will help some readers to clarify examples fo the term "gray literature" that are working papers, government documents, white papers and evaluations. Were legitimate government reports eliminated?
Author response: For this review, only original research works were considered. In that sense, any kind of report, summary, review, etc., independently of its origin, was not considered.
Author action: A couple of examples of “gray literature” work types have been added for clarification with the in-line
“[…] (e.g., government reports or white papers) […]”
- How was low-quality established? Publication impact factor? Other?
Author response: We thank the reviewer for making us aware of this point. We realized that the choice of words was not appropriate.
Author action: With that statement, we wanted to refer to some papers with severe formatting/editing issues, which made them unclear or unreadable. Since it is not related to impact factor or content, we decided to delete the statement for clarity. The corresponding sentence now reads
“Finally, the abstracts of the papers were reviewed, the key information was extracted from the main text, and those considered irrelevant were discarded.”
- It may be obvious to some readers that a crowdsourced data set is more of an open-access strategy relative to formalized clinical trial work, but the authors may wish to explain the data set as IRB-exempt participatory method of building a dataset.
Author response: We thank the reviewer for this suggestion.
Author action: A short clarification has been added at the end of section 3.1. pointing out this issue. It reads:
“Note that clinically obtained datasets usually require oversight from an Institutional Review Board, which takes time for the studies to run, data to be collected, and then shared. Hence, they less often exist. On the contrary, the applications developed to collect crowdsourced data include disclaimers informing the user beforehand that their audio recordings will be used for research purposes.”
- Figure 3 is unique and an excellent addition to the manuscript. Same comment for Figure 5, but Figure 4 was not clear; not sure if that is a font or page-size limitation.
Author response: We thank the reviewer for the good feeback of Figures 3 and 5. Figure 4 had a formatting problem that has been resolved.
Author action: Figure 4 has been reformatted and enlarged for easier reading and better understanding of its content.
- Table 1 reflects a significant amount of work, but could this be simply a data appendix available on-line? There were formatting issue on the PDF reviewed and the amount of data is overwhelming to include in the paper itself.
Author response: Despite of the large amount of data displayed on the table, we consider the table to be essential for the proper understanding of this work. The formatting issues in the previous submission made the table unreadble. This has now been resolved.
Author action: The table’s formatting has been corrected, the column “Subjects”, which did not offer information of relevance for this review, has been eliminated, and horizontal lines have been added for clarity.
- Lines 249-259 are informative and critical. Were databases segregated on method of data collection?
Author response: We thank the reviewer for the positive feedback. The method of data collection (crowdsourced or clinical) was mentioned in the Results section. It was, however, not considered during the discussion because the availability of detailed data (i.e., sampling rates, hardware, environment, methodology, etc.) was not enough to draw conclusions which might affect the performance of a classifier.
Author action: None.
- Lines 315-320 are important as they suggest a way to identify best-performing paper. Please consider expanding the narrative for clarity.
Author response: We thank the reviewer for this useful input.
Author action: As suggested, the narrative was expanded at this point to a deeper level of detail in the argumentation with the addition of
“ The entire diagnosis process includes several steps. Thus, there is a large number of variables involved. Each of the metrics describes a different aspect of the performance of a method. Therefore, the quality of a study improves when several metrics are reported since a high value on one metric does not directly imply high values in other ones. In this way, we considered those papers with three or more reported evaluation metrics of 90% or higher each to be amongst the best-performing. Note that other criteria could be used to determine the best-performing models. However, evaluating the study’s performance based on these metrics has to be consistently used in all papers.”
- The conclusion is somewhat anti-climatic. Possibly stronger recommendations based on the very comprehensive study? The reader is left without a clear sense of the authors' recommendation on what others might do (specifically) to improve acoustic detection of COVID-19.
Author response: We thank the reviewer for this suggestion.
Author action: A specific course of action expanding the already presented idea has been added to the conclusion for a better reference for future researchers, reading as follows:
“In particular, we suggest the individual study of every feature and the development of a quantifying method to establish the adequacy of each one for distinguishing between healthy and COVID-19-positive subjects. Although this might seem similar to dimensionality reduction methods, such as principal component analysis, an individual study of each feature would minimize the correlation in the feature selection step, thus strongly contributing to the reduction of error sources as well as to the generalization of the method.”
Furthermore, the following list of suggestions has been added at the end of the Discussion:
- “Database creators should aim to collect diverse data, including a wide range of ages, genders, ethnicities, etc. These datasets should also be publicly available since combining several databases would allow researchers to have a larger data corpus and would also reduce the possible biases introduced by demographic factors.”
- “ The pre-processing methods need to be automated and be non-data-specific as possible. It is essential that researchers consider carefully whether re-sampling techniques improve computational speed large enough to compensate for the information loss. An efficient CAD algorithm is critical since manual trimming and labeling become highly impractical when working with large datasets. Data augmentation should be avoided in favor of precise cough segmentation techniques and the creation of large datasets.
- “ The impact of each feature needs to be investigated individually, thus increasing the efficiency of manual feature extraction processes, allowing the adaptation of the neural network architectures, and improving the generalizability of the methods. Moreover, by fixing a subset of features, which does not depend on the available dataset, researchers could design studies to test the performance of several machine learning and deep learning models as the only dependent variable.”
Reviewer 2 Report
This paper presents a review of techniques used to detect COVID-19 in works in the last 10 years. The authors explained the background of its research but flaws in determine which is its real objective (hypothesis). The work is well written and ease to readers. The discussion about the used processing techniques was performed with several details. An important point to note is why this review is different from other similar works. The authors do not mention other reviews on the topic, such as:
- Buddhisha Udugama, Pranav Kadhiresan, Hannah N. Kozlowski, Ayden Malekjahani, Matthew Osborne, Vanessa Y. C. Li, Hongmin Chen, Samira Mubareka, Jonathan B. Gubbay, and Warren C. W. Chan. Diagnosing COVID-19: The Disease and Tools for Detection. ACS Nano 2020, 14, 4, 3822–3835.
- Serrurier, A.; Neuschaefer-Rube, C.; Röhrig, R. Past and Trends in Cough Sound Acquisition, Automatic Detection and Automatic Classification: A Comparative Review. Sensors 2022, 22, 2896. https://doi.org/10.3390/s22082896
- Schuller BW, Schuller DM, Qian K, Liu J, Zheng H and Li X (2021) COVID-19 and Computer Audition: An Overview on What Speech & Sound Analysis Could Contribute in the SARS-CoV-2 Corona Crisis. Front. Digit. Health 3:564906. doi: 10.3389/fdgth.2021.564906
- Thompson D., Lei Y. Mini review: Recent progress in RT-LAMP enabled COVID-19 detection. Sens. Actuators Rep. 2020;2:100017. doi: 10.1016/j.snr.2020.100017.
The authors should mention why their research are relevant in comparison with some similar works.
Major reviews:
- As there are many acronyms, I suggest the authors to perform a list at the end of paper.
- The introduction ends without establish the main objective and the hypothesis. It appears on Section 2, however, it needs to appear on section I. Moreover, why this research was performed? Why the importance to verify the methods of detection COVID-19 by audio signs? It should be highlighted on the text.
- The used time for the analyzed paper is questionable (linked with to the delimitation of research theme/motivation). If the papers were searched since 2012, the results could return only to cough and not for COVID-19, because the virus appeared on 2019. I suggest the authors to revise their objectives and motivations to this question be clearer.
- Figure 2 and 3 are not clear that are a histogram. It appears more than a bar graph of distribution. Please, revise them.
- For analysis of pre-processing: are there works, which used more than one pre-processing method?
- Table 1 have a lot of problems on this format, as in 3rd to 6th columns. Select the best feature that the authors want to compare. I advise to insert this table as supplementary file due to its size.
- The section of Refereces has no title.
Minor reviews:
- Lines 78 - 90: the author’s contribution should appear on the end of the text and not in the methodology process. Please, rewrite these sentences.
- Reference the work on line 136 on the “References” Section and not inside the text.
- Line 170: “Feature extraction” should be a subsection.
- Split the Figure 4 on separated figures. The legend is not fit on the page. Moreover, the axis titles are very small, difficult the reading. As there are several acronyms on “Frequency-space (F-type) features” last items, the authors can use one bar to identify these features, as “others”. The a) – d) items are not present on the Figure 4.
- Rebuild the Figure 6 because it presents the same problems of Figure 4.
- Please, improve the quality of Figure 7.
Author Response
This paper presents a review of techniques used to detect COVID-19 in works in the last 10 years. The authors explained the background of its research but flaws in determine which is its real objective (hypothesis). The work is well written and ease to readers. The discussion about the used processing techniques was performed with several details. An important point to note is why this review is different from other similar works. The authors do not mention other reviews on the topic. The authors should mention why their research are relevant in comparison with some similar works.
Author response: We thank the reviewer for the positive feedback and for this important recommendation.
Author action: The suggested works (not listed here) were examined and the focus of each one was mentioned in the text. The importance and new approaches of our work were then highlighted. The added text reads as follows:
“ Several papers have been published since the pandemic on this topic. Udugama et al. [15] and Thompson et al. [16] focus their reviews on clinical diagnostic methods. Serrurier et al. [17] presented a work that reviews automatic cough acquisition, detection, and classification methods. However, the included studies were not necessarily related to COVID-19. Schuller et al. [18] analyzed the applicability and limitations of computerized audio tools to contain the crisis due to the SARS-CoV-2 virus. A recent publication[19] discussed the challenges and opportunities of deep learning for cough-based COVID-19 diagnosis from an overall perspective. However, this review performs a detailed step-wise examination of the methodology of existing works, either using machine learning- or deep learning-based methods. Specifically, this work focuses on the audio pre-processing and feature extraction processes, which we believe will allow future researchers to improve the performance and generalization of these methods.”
MAJOR REVIEWS
As there are many acronyms, I suggest the authors to perform a list at the end of the paper.
Author response: We thank the reviewer for this suggestion.
Author action: An acronym list has been added at the end as an appendix.
The introduction ends without establishing the main objective and the hypothesis. It appears in Section 2. However, it needs to appear in section I. Moreover, why was this research performed? Why the importance to verify the methods of detection of COVID-19 by audio signs? It should be highlighted in the text.
Author response: We thank the reviewer for this correction.
Author action: The main objective and hypothesis were moved to section 1. The clarification of the importance of this review was also added (see “Author action” to the general comment of Reviewer #2)
The time used for the analyzed paper is questionable (linked with the delimitation of the research theme/motivation). If the papers were searched since 2012, the results could return only to cough and not to COVID-19 because the virus appeared in 2019. I suggest the authors revise their objectives and motivations to this question be clearer.
Author response: We thank the reviewer for this comment. We agree with this point of view. An initial 10-year-period in the search gave us a better overview of methods in automatic cough detection and the usage of the audio signals for diagnosing other respiratory pathologies. However, only papers published since December 1, 2019, were considered for this review, so we have corrected the time frame.
Author action: The corresponding sentence has been changed to
“To ensure the thoroughness of the review, several databases were searched for papers published between December 1, 2019, and January 1, 2022.”
Figures 2 and 3 are not clear that are a histogram. It appears more than a bar graph of distribution. Please, revise them.
Author response: We thank the reviewer for this suggestion. Both figures do depict categorical values and hence should be renamed as bar plots.
Author action: Both Figures, as well as Figure 5, have been renamed to bar plots.
For analysis of pre-processing: are there works that used more than one pre-processing method?
Author response: We thank the reviewer for pointing out this detail. Yes, most works used several pre-processing methods.
Author action: This has been clarified in the corresponding part of the Results. The sentence on lines 124-125 (old version) has been rewritten to
“While most papers employed several pre-processing techniques, three papers were found [31–33], which did not report using any pre-processing steps. Moreover, two other papers [34,35] explicitly specified working with the unprocessed signal.”
Table 1 has a lot of problems in this format, as in the 3rd to 6th columns. Select the best feature that the authors want to compare. I advise inserting this table as a supplementary file due to its size.
Author response: Despite the large amount of data displayed in the table, we consider the table to be essential for properly understanding this work. The formatting issues in the previous submission made the table unreadable. This has now been resolved.
Author Action: The table’s formatting has been corrected, the column “Subjects,” which did not offer information of relevance for this review, has been eliminated, and horizontal lines have been added for clarity.
The section of References has no title.
Author response: We thank the reviewer for pointing this detail out.
Author action: This issue has been resolved.
MINOR REVIEWS
Lines 78 - 90: the author’s contribution should appear on the end of the text and not in the methodology process. Please, rewrite these sentences.
Author response: We thank the reviewer for this suggestion.
Author action: These lines have been moved to the appropriate section.
Reference the work on line 136 in the “References” Section and not inside the text.
Author response: We thank the reviewer for this suggestion.
Author action: The work is now listed in the “References” section and the sentence has been rewritten as follows:
“[…], and created a cough segmentation algorithm [36] .”
Line 170: “Feature extraction” should be a subsection.
Author response: We thank the reviewer for this detail.
Author Action: The issue has been resolved.
Split Figure 4 on separated figures. The legend is not fit on the page. Moreover, the axis titles are very small, difficult the reading. As there are several acronyms on “Frequency-space (F-type) features” last items, the authors can use one bar to identify these features as “others”. The a) – d) items are not present in Figure 4.
Author response: We thank the reviewer for this input. However, we consider that Figure 4 should be one figure. We agree on the formatting and layout problem, so we have addressed these issues.
Author action: Figure 4 has been reformatted and enlarged for easier reading and a better understanding of its content.
Rebuild Figure 6 because it presents the same problems as Figure 4.
Asuthor response: We thank the reviewer for this suggestion. The same approach as in Figure 4 has been taken.
Author action: Figure 6 has been reformatted and enlarged for easier reading and better understanding of its content.
Please, improve the quality of Figure 7.
Author response: We thank the reviewer for this valuable input.
Author action: Figure 7 has been remade, and more details are added.